# Viral Appropriation of Specificity Protein 1 (Sp1): The Role of Sp1 in Human Retro- and DNA Viruses in Promoter Activation and Beyond

**DOI:** 10.3390/v17030295

**Published:** 2025-02-20

**Authors:** Kira Sviderskaia, Vanessa Meier-Stephenson

**Affiliations:** 1Department of Medicine, University of Alberta, Edmonton, AB T6G 2G3, Canada; svidersk@ualberta.ca; 2Department of Medical Microbiology and Immunology, University of Alberta, Edmonton, AB T6G 2R3, Canada; 3Li Ka Shing Institute of Virology, University of Alberta, Edmonton, AB T6G 2R3, Canada

**Keywords:** specificity protein 1 (Sp1), transcription factor, viral pathogenesis, retroviruses, DNA viruses, host–viral interactions

## Abstract

Specificity protein 1 (Sp1) is a highly ubiquitous transcription factor and one employed by numerous viruses to complete their life cycles. In this review, we start by summarizing the relationships between Sp1 function, DNA binding, and structural motifs. We then describe the role Sp1 plays in transcriptional activation of seven viral families, composed of human retro- and DNA viruses, with a focus on key promoter regions. Additionally, we discuss pathways in common across multiple viruses, highlighting the importance of the cell regulatory role of Sp1. We also describe Sp1-related epigenetic and protein post-translational modifications during viral infection and how they relate to Sp1 binding. Finally, with these insights in mind, we comment on the potential for Sp1-targeting therapies, such as repurposing drugs currently in use in the anti-cancer realm, and what limitations such agents would have as antivirals.

## 1. Introduction

Specificity protein 1 (Sp1) is a highly ubiquitous and evolutionarily conserved transcription factor (TF). Sp1 and its orthologs are found across eukaryotes, from yeast to higher-order mammals [1,2]. It is also found in essentially every cell type, expressed at varying levels (www.proteinatlas.org/ENSG00000185591-SP1/cell+line (accessed on 20 January 2025) [3,4]), and it has at least 12,000 binding sites in the human genome, binding to GC-rich motifs. Sp1 has been shown to be indispensable during embryonic development [5] and has been widely studied for its role in cancer biology because of its ability to transactivate genes involved in cell cycle regulation and immune system evasion, extensively reviewed elsewhere [6,7,8,9,10]. The Sp1 promoter itself is directly regulated by other transcription factors like Myc, Maz, and HIF, and it is autoregulated through several Sp1 binding sites [11]. The ubiquitous and multifunctional nature of Sp1 has opened opportunities for viral appropriation—many viruses use Sp1 for their own transcriptional activation. In this article, we aim to summarize the contemporary understanding of Sp1’s function and mechanisms as a transcriptional activator and more in human retro- and DNA viruses.

## 2. Sp1 Structure and Binding Motif

Sp1 belongs to the family of Specificity protein/Kruppel-like (Sp/KLF) transcription factors, which consists of 26 members, unified by the DNA domain made of three Cys2His2-type zinc fingers at the C-terminus. The Sp-like family is distinguished by the presence of a Buttonhead domain (BTD) and the Sp box at the N-terminus [12]. The former is said to participate in binding and transactivation [13], while the latter has an endolytic cleavage site for its presumed proteasome-dependent proteolysis, involved in cell cycle control [14]. The Sp1–4 proteins are further separated from the other five Sp1-like protein members by the presence of glutamine-rich regions (Figure 1A). Of them, Sp3 protein is the most closely related to Sp1, binding to the same motif, but it is reported to have repressive effects on transcription initiation [15,16,17,18].

Sp1 contains two transactivation domains, A and B, which cooperate with a highly charged DNA binding domain, C. These domains support DNA binding and transactivation of Domain D, which is responsible for its proposed tetrameric multimerization, capable of bridging cis- and trans- Sp1 binding sites (Sp1bs) (Figure 1A) [12,23,24]. Detailed information on the nature of this multimerization is currently lacking, as studies are limited. The DNA-binding domain contains three zinc ‘fingers’ formed by β-turns with an α-helix stabilized at the base by zinc ions (Zn^2+^) (Figure 1C). Due to the intrinsically disordered nature of Sp1, only the DNA-binding zinc finger domains have been structurally elucidated via NMR [10,19,25] (Figure 1B). It is hypothesized that the flexibility of the Sp1 structure allows it to interact with a large number of cellular partners in a “coupled folding and binding” fashion, a state that would allow disordered sections to become more ordered upon binding. However, these conformation-dependent interactions have yet to be better characterized (see Transcription Initiation section). Lastly, Sp1 contains an inhibitory domain (ID), identified based on sequence similarity, most of which was shown to be excised under cellular stress just prior to the cleavage site within the Sp box (Figure 1A) [26]. To the best of our knowledge, direct Sp1-led transcription inhibition via the ID domain is yet to be demonstrated and therefore will not be discussed further. Descriptions of ID and other domains are available at: https://www.uniprot.org/uniprotkb/P08047/entry (accessed 20 January 2025) [20,27]. Overall, Sp1’s actions are executed through direct binding to DNA, most commonly to promoter regions, where it interacts with the basal transcription machinery, other TFs, and cell regulators [28].

### 2.1. Nucleic Acid Binding Motifs

Zinc fingers of Sp1 and related proteins wrap around the helical DNA by binding into its major groove [10,29]. Each zinc finger has its own preferred DNA binding sequence, typically 3 nucleotides (nt), and when combined, it can recognize segments of up to 9 nt [21,30,31]. The human Sp1 binding motif consists of several conserved guanine (G) residues separated by a single other base, commonly cytosine (C), and sometimes thymine (T) [32] or adenine (A). Some other vertebrates are, however, more likely to have variation in the motif, like with a GA-box instead. It is hypothesized that eutherian mammals and birds have independently evolved characteristic GC-boxes as Sp1 binding sites. Accordingly, the ancestral genes that had a GA box are less likely to have a methylated GC-box (see Epigenetics in Viral Infections) in contemporary versions, pointing to the conserved transcription activation role [2]. Lastly, there are variations of GC-, GA-, and GT-boxes in promoters of various human papilloma viruses, which may also stem from evolutionary pressures [33]. Proposed variations of the Sp1 binding motif are available at: https://jaspar.uio.no/matrix/MA0079/versions (accessed 20 January 2025) [22,34] (Figure 1D). Beyond its conventional role as a promoter activator, Sp1 is now recognized to contribute to mRNA stability via direct binding to AG-rich RNA sequences. The zinc fingers are again presumed to bind to mRNA, with a lower affinity compared to their DNA counterpart (Figure 2A) [35].

It is well characterized that multiple Sp1bs in promoters grant increased activation opportunities for the gene ahead [36,37]. Thus, the “superactivation” or “multimerisation” hypothesis, where several close neighboring binding sites allow Sp1 molecules to interact with each other as they are bound to the promoter (Figure 2C), emerged, although in vitro evidence is still lacking. An additional explanation for non-conventional secondary structures formed by several Sp1bs and other GC-rich motifs has started to gain recognition. Guanine-quadruplexes (G4Qs) are non-canonical forms of DNA or RNA whereby G-rich regions can create planar arrangements of G’s from Hoogsteen bonds (as compared to the canonical Watson–Crick base-pairing), known to bind some zinc fingers, including Sp1 (Figure 2A) [38,39]. Positioned in the promoters, these structures act as TFs anchors and are indispensable for transcription initiation [40]. We do not exclude that some of the Sp1bs discussed below also form G4Q structures that are yet to be reported.

### 2.2. Transcription Initiation

Transcription executed by human RNA polymerase II (PolII) starts with a pre-initiation complex (PIC) formation. According to the classical model, TFIID, a multi-protein subunit complex, brings a TATA-binding protein (TBP) onto the TATA box with the help of a TFIIA dimer. Following that, TFIIB, Pol II-TFIIF, and mediator complexes are recruited to induce structural rearrangement, leading to strand separation and subsequent transcription [41].

The exact structural interaction between Sp1, the DNA strand, and the PIC are yet to be elucidated; however, sequence homology identified a 9aaTAD-like motif, commonly present in transactivator proteins and interacting with TBP-associated factor 9 (TAF9) [42]. Moreover, the same motif was shown to interact with TFIID-associated factor 4 (TAF4) via their respective highly disorder glutamine (Q)-rich domains. Determined through mutagenesis studies, Sp1’s interaction domain is a stretch of hydrophobic residues (464-WATLQLQNL-472) (Figure 1A) [43]. Interestingly, using NMR, no significant conformational change associated with the Sp1-TAF4 interaction was reported—an unexpected finding that goes against the well-accepted concept of “coupled folding and binding” for intrinsically disordered proteins [43].

### 2.3. Binding Kinetics

In a context-dependent manner, Sp1 binding affinity relies not only on the site’s nucleotide composition but also the location. Within the human genome, the binding of Sp1 was characterized to be either “fast” or “slow”. “Fast” binding sites had higher motif strength, with sequence composition closer to the JASPAR MA0079.3 consensus. They were found in promoters and regions containing several Sp1bs, in accordance with a concept of “superactivation”, which argues for an increase in promoter activation associated with multiple Sp1bs (see Nucleic Acid Binding Motifs). Conversely, “slow” binding was found in enhancers and polycomb-repressed regions. The functional significance of these differences is yet to be described, as a direct link was not found between the binding speed and the nucleosome occupancy or higher transcriptional activity [37].

### 2.4. Function

As noted above, Sp1 acts as a transcription factor and aids in both the formation and activation of the replisome needed for transcription to initiate. Like other TFs, the action can be executed by a variety of mechanisms [44]. The omnipresence of Sp1 binding sites in gene promoters implicates it in various maladaptive processes like the activation of viral genes [17,45,46,47,48,49,50] or oncogenes [7,8,10,46].

In many cases, Sp1 is required to bind to viral genomes to initiate transcription upon viral entry; therefore, many viral pathways rely on an increase in Sp1 availability. Yet, Sp1 also activates cellular transcription, including that of immune genes, like those involved in the RIG-I pathway [51,52], and accordingly, viruses can promote their latency by silencing Sp1-directed transcription [53].

In some viruses like those with bidirectional promoters, Sp1 also participates in viral phase control through differential binding affinities to sites in early and late promoters by competing with other cellular TFs and viral transactivators for the same sites. It can also influence transcriptional activity via chromatin remodeling through impacts on histone acetylation and/or DNA methylation (see Epigenetics in Viral Infections). Firstly, Sp1 bound to a promoter protects the viral genome from methylation [54,55,56,57,58,59]. However, Sp1 also cooperates with opposing histone modification enzymes—histone acetyltransferase (HAT) and histone deacetylase (HDAC)—that promote and silence transcription (Figure 2D,E). Overall, Sp1’s impact on transcriptional activation and how this relates to the viral life cycle is multifaceted and complex.

In addition to its direct impact on expression, Sp1 may tether another protein to the promoter, acting as an intermediate between DNA and a protein that lacks a binding site (Figure 2D) [57,60]. This increases the multitude of Sp1-directed pathways to the nth degree; a loophole commonly exploited by viruses. In some cases, Sp1 is proposed to form a dimer with another protein, where each molecule is bound to a separate promoter to bring the replisome complex to activate transcription (Figure 2F) [61]. Lastly, Sp1 has been reported to influence mRNA stability in a tissue- and context-specific manner via alternative polyadenylation (APA) by directly binding to AG-rich RNA, similarly to other zinc-finger proteins (Figure 2A) [35].

## 3. Sp1 Use by Human Viruses

### 3.1. Retroviruses

Retroviruses are a diverse family of +ssRNA viruses that lead to immune system impairments, chronic inflammation, and oncogenesis. They exist as both exogenous viruses and endogenous viruses, passed down as part of the host’s genome [62]. These viruses reverse transcribe and insert themselves into the host genome, where they rely on host machinery for completion of their replication cycle. Discussed here are lentiviruses (HIV-1), delta-retroviruses (HTLV), and beta-retroviruses (HERV-K) (see also Table 1).

#### 3.1.1. Lentiviruses

Human Immunodeficiency Virus-1 (HIV-1 is the underlying viral cause of acquired immunodeficiency virus syndrome (AIDS): An estimated 39.9 million people worldwide were living with HIV in 2023, and of those, 29.8 million people were on antiviral treatments that undoubtedly turned a lethal disease into a manageable chronic condition [119].

HIV-1 has a linear bidirectional RNA genome ~9.4 kb in length, packaged with its reverse transcriptase, integrase, and protease. Upon integration into the host genome, cellular TFs facilitate low levels of HIV-1 transcriptional activation that allows for Tat protein synthesis. RNA PolII is recruited to the HIV-1 promoter, and transcription is initiated then stalled shortly thereafter to produce a short RNA sequence that folds into transcription responsive elements (TARs). Tat protein, upon accumulation, binds the nascent TAR RNA, along with other associated factors, including positive transcription elongation factor (p-TEFb), releasing the stalled RNA PolII and enabling full genome transcription. This ultimately leads to an increase in transcription in a positive-feedback fashion [120].

The viral promoter LTR consists of a TATA box, two nuclear factor kappa-light-chain-enhancer of activated B cells (NF-κB), and three Sp1 binding sites, as well as other regulatory elements, which together permit transcription of the full genome (Figure 3A, top). Four mutually exclusive G4Qs are reported to be formed by three Sp1bs in the 5′LTR [39]. Downstream of the transcription start site (TSS), there is another set of TF binding sites, including one or two Sp1bs, depending on the strain [68].

Nucleotide variations in the LTR were shown to impact HIV-1 replication kinetics and virulence. For example, an extra NF-κB binding site or a single-point C to T mutation in the Sp1bs-III of subtype C correlated with increased transcriptional activation and is hypothesized to induce more successful viral expansion [63,64,65]. Meanwhile, other mutations, like in the 4th guanosine of the same Sp1bs-III, disrupted viral transcription when introduced through site-directed mutagenesis [64]. In a similar model, single-point mutations in the TSS in specific TF binding sites did not yield a reduction in transcriptional activation, while simultaneous point-mutations in Sp1, activation protein (AP)-1, AP-3-like, and DRE binding factor (DBF) 1 in TSS fully inactivated the LTR [68].

Sp1’s highly integral role in promoter activation places it at the forefront of involvement with many cell types susceptible to HIV-1 infection (T cells, monocytes/macrophages, iDC, and microglia) [16]. For instance, in vitro evidence in T lymphocytes shows that Sp1 contributes to HAT and HDAC that enhance and reduce transcription, respectively [123]. The recruitment of HDAC to LTR through Sp1 is also executed by an innate immune factor human tripartite motif protein 5α (TRIM5α) as a way to restrict reverse transcription [71]. Additionally, γ-IFN-inducible protein 16 (IFI16), an antiviral factor, binds Sp1 protein, reducing viral replication [73].

In microglial cells, Sp1 was reported to act as an anchor for nuclear factor for interleukin 6 (NF-IL6), cAMP-response element binding protein (CREB) and chicken ovalbumin upstream promoter (COUP)-TF to increase NF-κB levels, leading to Tat expression and supporting transcriptional activation [18]. Conversely, Sp1bs can also anchor COUP-TF-interacting protein 2 (CTIP2) and lysine-specific histone demethylase 1 (LSD1), triggering deacetylation and viral latency [72]. Additionally, microglial cells express a related protein, Sp3, that represses Sp1 activity by blocking the binding site [23], analogous to what was reported to occur in monocytes [62].

HIV-1 LTR promoters have the ability to switch between silent and active states, which contributes to the maintenance of the latent viral reservoir—an integral part of infection persistence. Sp1 functions as an intermediate protein to bring other TFs and co-factors to the promoter to create the replisome for viral expression. When Sp1 binding sites are mutated, the Tat-mediated positive-feedback loop fails to be established, leading to stochastic changes in the transcriptional phenotype in infected cells, resulting in poor establishment of infection [66,67].

#### 3.1.2. Delta-Retroviruses

Human T-lymphotropic virus-1 (HTLV-1) infects upwards of 10 million people globally, although the numbers are almost certainly an underestimation [124,125]. Most chronic HTLV-1 carriers are asymptomatic; however, for some, there is a risk of an aggressive adult T-cell leukemia (ATL; ~5% lifetime risk) or an HTLV-1-associated myelopathy/tropical spastic paraparesis (HAM/TSP; 0.18 to 1.8% lifetime risk) [125].

The HTLV-1 retrovirus structurally consists of a linear bidirectional 8.7 kb linear RNA genome, packaged with the same three enzymes as HIV-1 (reverse transcriptase, integrase, and protease). Similarly, the genome contains two LTRs which act as cis-regulatory elements, and upon integration into the host genome, the provirus transcribes a segment of its RNA, known as the tax/rex RNA. Tax and rex proteins will work to jumpstart the infection [66] and allow unspliced RNA to exit into cytoplasm, respectively. Another regulatory gene is HTLV-1 bZIP factor (Hbz), which promotes leukemogenesis, latency, and non-immunogenicity, interfering with Tax-mediated pathways and acting on host genes [46].

There are six identified Sp1bs in HTLV-1’s genome distributed in sets of two in the U5, R, and U3 regions of each LTR. The sense strand, controlled by the 5′LTR, encodes structural genes and tax/rex mRNA. The 5′LTR includes three Tax responsive elements (TREs), two Sp1bs, and a TATA box (Figure 3A, bottom) [46]. To activate transcription, the CREB/activating transcription factor (CREB-ATF) complex binds viral cyclic AMP-response elements (CREs) in the 5′LTR, along with CBP/p300 protein, which acetylates the region, making it accessible for transcriptional complexes to bind, including Sp1 and the viral polymerase [57]. The 3′LTR, controlling the HBZ “latency protein” coded in the antisense direction, includes two Sp1bs and lacks a functional TATA box [46]. HBZ induces its expression in a positive feedback loop by forming an HBZ–JunD complex with an Sp1 protein bound to the 3′LTR. The HBZ–JunD–Sp1 complex is also able to trigger expression from the human telomerase reverse transcriptase (hTERT) promoter, stimulating cell immortality [57].

To interfere with Tax production, latency-promoting HBZ mRNA is hypothesized to bind to U5 and displace TBP binding, thereby repressing subsequent transcription [66,124]. However, mutations in the HBZ mRNA stem loop did not interfere with HTLV-1 infection, which suggests that the function is performed by the HBZ protein [126]. HBZ mRNA was also reported to modulate host chemokine receptor type 4 (CCR 4) expression, contributing to ATL. HBZ protein also interferes with Tax expression by preventing CREB-2 from binding to CRE at U5, preventing Sp1 protein and polymerase binding [57]. Lastly, HBZ protein was reported to counteract Tax-mediated NF-κB upregulation by binding to ATF/AP-1 transcription factors [125].

The current understanding of HTLV-1 latency with regards to epigenetic modifications is limited; however, the 5′LTR, including Sp1bs, was reported to be hypermethylated when integrated [75], and accordingly, Tax is not expressed in approximately half of ATL cases [57]. Suppression of Tax expression is proposed to hide the virus from the immune response. To date, it is not known how the infection persists and what conditions trigger Tax expression and subsequent virus re-activation. However, upon expression, Tax-directed LTR re-transcription is suggested to be executed via Tax forming a complex with methyl-CpG binding domain 2 (MBD2) protein that binds to CRE elements of highly methylated promoters with high affinity [75]. Conversely, the 3′LTR remains intact in all ATL cases, underlining the importance of HBZ for leukemogenesis [57].

#### 3.1.3. Beta-Retroviruses

Human endogenous retrovirus (HERV)-Ks are human-endogenous viruses that have been incorporated into germ cell DNA, but upon generations of mutations, have lost their ability to form a complete virion. Some sequences maintain an open reading frame and are capable of producing retrovirus-like particles but are usually epigenetically silenced. The impact of HERVs is ambivalent, with the ability to increase pathogenesis in viral infection, carcinogenesis, and immune disease, and to decrease disease progression through immune activation. It is estimated that HERVs comprise up to 8% of the human genome, with 90% of them being solo-LTR [127].

Full-length HERV-K are ~9.5 kb, but most of them contain various deletions [128]. HERV-K LTR are controlled by cell-specific TFs like hypoxia-inducible factors (HIFs), NF-κB, initiator element (Inr), Yin Yang 1 (YY1), Sp1, and Sp3 [129]. Sp1 is known to participate in chromatin remodeling and protection from methylation of GC boxes, especially during early embryogenesis in germ cells, where endogenous viruses replicate to disseminate within the genome. Therefore, it is likely that Sp1 plays an integral role in HERV-K genome propagation, as many other TFs are not present in undifferentiated cells [55]. Moreover, Sp1 overexpression in cancers is suggested to support HERV-K promoter demethylation, leading to the production of viral proteins that further contribute to pathogenicity [76].

### 3.2. DNA Viruses

Sp1’s use by DNA viruses are discussed for Hepadnaviridae, Papillomaviridae, Herpesviridae, Polyomaviridae, Poxviridae, Adenoviridae and Parvoviridae (See also Table 1).

#### 3.2.1. Hepadnaviridae

Hepatitis B virus (HBV) is estimated to infect upwards of 300 million people worldwide [130,131]. Chronic HBV infection can lead to liver fibrosis, cirrhosis, and liver failure. HBV is heavily linked to liver cancer, for which it was recorded to have caused 265,000 fatalities in 2015 alone worldwide [132].

HBV exists as a circular partially dsDNA virus and is remarkably compact at only ~3.2 kb. HBV genome expression is regulated by the four promoters—surface (S) 1, PreS/S2, X, and PreCore/Core (C) promoters, as well as two enhancer (En) regions directing four overlapping reading frames. For the scope of this article, notable proteins include Core, a nucleocapsid protein that participates in viral replication, and X (or HBx), an immune system-suppressing protein that alters the epigenetic state and increases oxidative stress, leading to cell immortalization [133].

Viral expression is controlled by a multitude of host transcription factors that can be divided into liver-enriched and ubiquitous groups [134]. Belonging to the latter group, Sp1 plays a central role in HBV replication, where all but the X promoter have Sp1bs [39,48]. The S1 promoter contains one Sp1 site overlapped with a TATA box [135]. The PreS/S2 promoter contains at least three Sp1bs shown to control surface protein expression [82,135]. The PreC/C promoter contains two neighboring Sp1bs, capable of forming a G-quadruplex, contributing to core protein expression [38,47]. Mutations impacting the quadruplex lead to loss of the downstream mRNA and core protein [39,77]. The Sp1bs present in enhancer II (EnII) contribute to expression from all four promoters (Figure 4A) [77].

Sp1 protein interacts with other transcription factors including Krüppel-like factor 15 (KLF15) protein, a binding site overlapping with Sp1bs in the Core and PreS2/S2 promoters, allowing for potential positive synergistic effects [78]. Conversely, Sp1 forms a complex with NF-κB, which prevents DNA binding of either protein to preC/C and EnII, leading to inhibition of protein expression, although a similar action could happen at the S1 and PreS2/S2 promoters [79]. HBx transformative protein was reported to downregulate NF-κB, which prevents immune system recognition and increases the availability of Sp1, contributing to hepatocellular carcinoma (HCC) formation [80,84]. Thereafter, increased availability of Sp1 activates the expression of immortality-related genes like hTERT [138], tumor suppressors like the Dickkopf WNT signaling pathway inhibitor (DKK) 1 [139], or other transcription factors like the NF-κB itself. On the other hand, HBx can also upregulate NF-κB, which activates pro-survival genes, once again leading to HCC [84].

During HBV infection, the HBx protein changes a cell’s epigenetic profile considerably to allow for productive chronic infection. For example, HBx increases the activity of DNA methyltransferases (DNMTs) and hence downregulates tumor suppressor genes like Rat sarcoma (RAS) association domain family member 1A (RASSF1A) [83] and cyclin-dependent protein kinases (CDKs) involved in cell cycle control [75]. Overall, complex Hbx-induced methylation changes are still poorly understood [75,81]. HBx enhances Sp1 expression, and by extension, the expression of its own genes through an interaction with long non-coding RNA, like lncRNA homeobox (HOX) transcript antisense RNA (HOTAIR), which has a positive binding site on the Sp1 gene promoter [78]. Lastly, HBx also forms a complex with HDAC1 to deacetylate the Sp1 protein, reducing its DNA binding affinity and leading to the transcriptional repression of Sp1-dependent genes [80].

#### 3.2.2. Papillomaviridae

Human papilloma viruses (HPVs) also can be divided into mucosal, also referred to as genital, and cutaneous HPVs. Mucosal HPVs are mostly composed of alpha-HPVs that include high-risk carcinogenic types (HPV-16/-18), while cutaneous HPVs cause benign papillomas or warts [48]. In 2018, around 700,000 people developed HPV-associated cancers, with 90% of them being women [140].

Papillomaviruses (PVs) are circular dsDNA viruses, ~8 kb in length [136]. Their viral life cycle is loosely divided into three stages: initial amplification, early expression, and late expression. Upon infection, E2 loads viral helicase, E1, onto the viral upstream regulatory/control region (URR/LCR) for the formation of a replisome [54]. Collectively, URR is bound by a broad range of cellular transcription factors, including YY1, AP1, neurofibromatosis type 1 (NF1), octamer 1 (OCT1), NF-IL6, keratinocyte-specific transcription factor 1 (KRF1), NF-κB, Sp1, etc. (Figure 4B) [86,137,141]. URR directs viral expression to be either early or late. Early promoter activation ultimately contributes to carcinogenesis via the transformative properties of E6 and E7, which participate in the establishment of persistent infection and subsequent cell transformation, as extensively formulated in mucosal HPVs.

Most mucosal HPVs’ URRs have four E2 binding sites (E2bs), three of which are in the proximal promoter, along with the Sp1 binding site and TATA box. Transcriptional activity of the early promoter is controlled by an antagonistic relationship between Sp1 and E2 proteins. Specifically, E2 binding to these three E2bs creates steric hindrance and represses viral transcription via exclusion of Sp1 and other transcription factors that would have otherwise contributed to replisome formation, including TFIID [137], TFIIB [142], and TBP [54]. Conversely, early protein transcription is activated upon E2 binding to the distal site, which has a higher binding affinity for E2 than the more proximal viral-repressing sites. Thus, in the event of E2 scarcity, such as at the onset of infection, the early promoter is upregulated, while it is downregulated in E2 abundance (Figure 2B) [85]. Moreover, HPV integration-associated loss of the E2 gene leads to lack of an antagonistic relationship between E2 and Sp1, contributing to dysregulation of E6/E7 protein expression [48]. Cancer progression in patients correlates with increasing DNA methylation of the proximal vs. distal E2bs, in accordance with E6/E7 transcriptional activation. Yet, methylation at the proximal site was also found to be a positive prognostic factor in HPV-16 [85]. Of note, HPV integration into the host genome occurs in a variety of ways in carcinogenic environments [143], and the transforming properties of HPV extend beyond just the Sp1–E2 relationship (Figure 4B) [144].

The HPV-16/-18 genomes have two nucleosomes located in the URR region. Cell differentiation results in chromatin opening around both early and late promoters, attributable to increased transcription factor binding and to histone modifications. Sp1 can displace nucleosomes and relieve transcription repression in HPV-16 in vitro [59], while cell differentiation executes histone acetylation and DNA methylation, changing TF affinity. In undifferentiated cells, Sp1 has a 5-fold greater affinity for the late compared to the early promoter. Upon differentiation, affinity for the late promoter stays approximately the same, while binding to the early promoter becomes 5-fold stronger than to the late promoter [86].

Among all alpha-PVs, the early promoter region of URR is conserved, including the presence of the Sp1 site [137]. Naturally occurring point mutations in Sp1 motifs were reported to impact binding and promoter activity in vitro [145]. However, the exact Sp1 motif is not conserved; for example, in HPV16, the Sp1 motif is a GC box, while in HPV 18, it is a GA box, and in HPV 45, it is a GT box [33]. Since cutaneous HPVs predominantly have a different genome structure, the Sp1 site cannot be identified in some. Adding Sp1bs in their promoter did yield strong activation, likely due to the comparatively weaker URR enhancer [90,146].

#### 3.2.3. Herpesviridae

Herpesviruses are a diverse group of linear dsDNA viruses, subdivided into alpha-, beta-, and gamma-herpesviruses based on their tissue tropisms. Alpha-herpesviruses infect epithelial and neuronal cells, while beta- and gamma-herpesviruses infect a wide variety of immune cells or are restricted to B cells, respectively [147,148]. Their impact on the human host varies depending on the virus, site of infection, and host’s control of the virus.

The genome size of Herpesviridae varies from 120 to 230 kb. Herpesviruses’ life cycles are separated into two distinct phases: latent and lytic. These viruses are defined by their ability to establish latent infection soon after entry, allowing them to avoid the immune system for years. However, to maintain an infectious reservoir for transmission, they may be stimulated into the lytic stage. All viral proteins are divided into the following classes: immediate-early proteins that act as transactivators; early proteins that are required for genome replication; leaky late proteins, expressed prior to and upregulated post-DNA synthesis; and true late proteins that are expressed exclusively after viral synthesis. The late proteins are required for assembly and egress [149].

##### Alphaherpesviruses

Herpes simplex virus-1 (HSV-1) has a tropism for epithelial and sensory neuronal cells; in the latter, it is able to establish latent infection [91] and may lead to encephalitis, blindness, and neonatal deformities in rare cases [58]. It is estimated that 3.8 million people are infected worldwide [150].

The HSV virion contains a single linear dsDNA genome of 152 kb. Upon entry of the HSV-1 virion, Sp1 binds within promoters of most immediate early (IE) and early (E) genes. Sp1 was shown to be essential for expression of IE genes like infected cell protein 0 (ICP0), ICP4, and ICP27 [87]. IE gene promoters possess the most binding sites and therefore require the most TFs to be activated, while late gene promoters can be activated merely with a TBP protein. The drive behind such conservation is yet to be determined [151]. ChIP-Seq showed largely matching Sp1 and Pol II binding to most IE and E gene promoters 2 h post-infection (hpi). However, 4 hpi, a drastic reduction in Sp1 binding and Pol II in Sp1-bound promoters was recorded, alluding to the link and the integral role of Sp1-mediated transcriptional activation in HSV-1 [88]. The change in promoter occupation is suggested to be caused by Sp1 phosphorylation, leading to reduced ability to transactivate [87], or an increase in viral copies, leading to relative decrease in Sp1 concentration [88].

As viral transactivators are being expressed, Sp1 becomes redundant in the context of transcriptional activation. However, in the absence of viral transactivators in the latent stage, or under IFN-induced stress [45], as demonstrated in human foreskin fibroblasts, Sp1 once again becomes important to support basal transcription levels. As well, in Vero cells, a stress-induced glucocorticoid receptor TF forms a complex with KLF15, which transactivates IE genes by binding at Sp1bs and displacing Sp1 protein (Figure 2E) [89]. Whether the complex can truly displace Sp1 from the promoter or not in vivo is yet to be shown. Additionally, Sp1 was shown to be indispensable for viral protein 5 (VP5) expression, a leaky-late gene [88,90].

VP16, a late gene, presents an interesting case, where simultaneous mutations in three Sp1bs before the TATA box did not yield a reduction in transcriptional activity, rendering the sites redundant [91,92]. Instead, within the gene, there is an early growth response protein -1 (Erg-1)/Sp1 binding sequence that is suggested to play both positive and negative regulatory functions. A stress-induced factor, Erg-1, is hypothesized to bind to the site and displace positive-regulating Sp1 protein. Erg-1 also brings the NGFI-A-binding protein 2 (NAB2), a co-repressor, to induce chromatin closure, preserving latency during stress. So far, the binding of Erg-1 and Sp1 proteins to the region has been confirmed, but since the sequence is capable of forming a G4-quadruplex (G4Q), it may bind other TFs. A similar region is also found in the VP16 promoter in HSV-2 [58].

##### Betaherpesviruses

Human cytomegalovirus (CMV) or HHV-5 is a beta-herpesvirus that infects from 44 to 96% of the population depending on the region. It is most known for its association with birth defects and morbidities in immunocompromised individuals [96].

CMV virions contain a linear, dsDNA genome of ~235 kb [96]. Its transactivators are transcribed following the MEI promoter (MEIP), which includes binding sites for host transcription factors like NF-κB, CREB, AP-1, NF1, etc. [152]. These include seven Sp1bs in the promoter enhancers, of which at least two sites in the proximal enhancer were deduced to be redundant; hence, transcription was inhibited only upon mutation of both sites [94]. Nonetheless, Sp1 is upregulated upon binding of the CMV cell surface proteins glycoprotein (g) B and hemoglobin (h) B to the cell receptors during the course of lytic infection [97,98]. Sp1 also interacts with IE72 (or IE1) and IE86 (or IE2), well-described CMV transactivators, to activate MIEP and other promoters [95]. IE72 specifically lacks a DNA-binding domain, and it is proposed to use Sp1 to tether to its active site and maintain the genome during latency (Figure 2D) [96].

##### Gammaherpesviruses

Epstein–Barr virus (EBV) or HHV-4 is a gamma-herpesvirus that infects over 90% of the population around the world. Lytic infection of EBV commonly causes mononucleosis, while latent infection is associated with a variety of cancers [148].

The EBV virion contains a linear, dsDNA genome of ~180 kb. Sp1 is involved in EBV activation at several points: firstly, EBV transactivator genes, Zta and Rta that synergistically activate each other and other genes, both have two and three Sp1bs in their promoters, respectively (Figure 3B). However, Zta promoter (Zp) was reported to only require Sp1 for full (as opposed to leaky) expression [100]. Moreover, Zp and some other genes lack Rta-response elements. Presumably, Sp1 acts as an intermediary in the Sp1—MBD1-containing chromatin-associated factor 1 (MCAF1)—Rta complex to grant Rta activation properties without direct binding (Figure 2D) [60].

During latency, the Zp promoter is repressed by HDAC2. It is hypothesized that phosphorylated Sp1 releases HDAC2 from the three Sp1bs in the ZID element of Zp [101]. At the same time, p53 protein forms complexes with TFs including Sp1 and indirectly binds to Zp [99], activating transcription. Hau et al. reported that ATM-mediated cellular DNA damage response contributes to the lytic stage via phosphorylation of Sp1, leading to increased viral replication in epithelial cells [102].

#### 3.2.4. Polyomaviridae

Polyomaviruses are dsDNA viruses that are divided into six genera, three of which infect humans, causing cytopathic, immune, and oncogenic pathologies [153]. The most common species of polyomaviruses are Merkel cell polyomavirus (MCPyV; Alphapolyomavirus), the only oncogenic polyomavirus to date that causes fatal skin cancer, BKPyV (Betapolyomavirus), the causal agent for nephropathy, which is the leading cause for kidney transplant failure, and JCPyV (Betapolyomavirus), which is associated with a fatal demyelinating disease, progressive multifocal leukoencephalopathy (PML) [56]. The seroprevalence of MCPyV is up to 90% [154]. These viruses cause disease almost exclusively in immunocompromised individuals.

Polyomaviruses are circular dsDNA viruses, ~5 kb in size. Their genomes are made up of early and late genes controlled by a bidirectional promoter/enhancer region containing the origin of replication called the non-coding control/regulatory region (NCCR/NCRR) [56]. The NCCR can bind a variety of TFs; host factors like Sp1, NF1, Ets1, and a viral transactivator large tumor antigen (LTag) were found to have the most impact on expression (Figure 4C) [49,103,104]. Based on various predictions in silico, every human polyomavirus may have at least 1 and up to 14 Sp1bs in their NCCR [105,106]. Importantly, not all predicted sites bind Sp1 protein in vivo. Multiple Sp1bs are also suggested to protect the NCCR from silencing via host methylation [56]. For example, a pathogenic strain of JCV, Mad-1 isolate, has Sp1bs within the Sp1/Erg-1 repeat sequence, which was shown to bind only Erg-1 but not Sp1. Yet, JCV isolates have other Sp1bs and are most likely controlled via Sp1 expression [155].

Composition of NCCR has been linked to disease severity: pathogenic patient strains favored early gene expression compared to the archetypes found in immunocompetent hosts with asymptomatic infections. In the BKPyV archetype, the scale is tilted towards late expression through higher binding affinity of the Sp1bs proximal to the start of late genes compared to the site next to early genes [103]. In mutated strains where Sp1 or Ets binding sites proximal to late gene start are deleted, or an extra NF1 adjacent to early gene start is added, shifting of expression towards early—and therefore lytic—genes can occur. Additionally, deletion of two Sp1 sites, proximal to early gene start, led to complete abrogation of viral replication—such sequences have not been observed in clinical samples (Figure 4C) [49].

#### 3.2.5. Poxviridae

Poxviridae is a highly diverse family of large linear dsDNA viruses. The most-studied members include variola virus, the causative agent of smallpox epidemics, and vaccinia virus, the laboratory-passaged strains used for smallpox eradication in the past and as a platform for therapeutics delivery today. Lastly, the Mpox viruses are a re-emerging pathogen that infected over 100,000 people in the 2022–2024 period [156].

Poxviruses’ genomes range from 130 to 230 kb. These viruses are the only DNA virus family in which the virion never enters the nucleus of the host cell. As a result, the viruses are forced to encode their own transcriptional machinery and replicate in viral inclusion bodies [107]. However, the Sp1 protein, along with YY1, Pol II, and TBP, have been suggested to leak from the nucleus and associate with the genomes of vaccinia viruses [108]. To the best of our knowledge, it is unknown whether this association contributes to viral transcription or is merely a consequence of the DNA affinity of the host proteins.

#### 3.2.6. Adenoviridae

Human adenoviruses (HAdVs) are subdivided into species (A–G) and display various tissue tropisms that are capable of causing respiratory, ophthalmic, gastrointestinal, and neurological infections [157]. Adenoviruses cause most infections in children under 5 years old due to lack of humoral immunity. Seroprevalence varies vastly across the world and is well summarized here [158].

Adenoviruses (AdVs) are large, non-enveloped dsDNA viruses of ~26–46 kb. Transcription is controlled by viral proteins, including different forms of early (E) proteins E1A and E1B, and host factors including TFIID, cAMP, NF-κB, and Sp1 [50]. Sp1bs are present in the inverted terminal repeats (ITR) of HAdV-4 (species E), HAdV-7 (B), HAdV-40/41 (F), and HAdV-52 (G) due to a high conservation of the region within the subfamily of Mastadenoviruses [109,110]. Sp1 was reported to be a transcriptional activator of E1B [111], protein IX [112], in promoters of AdVs-2 and 5 (species C). Sp1 also activates the major late promoter (MLP) in AdV-4 (species E) and HAdV-7 (B) by forming a complex with Myc-associated zinc finger (Maz). Conversely, Sp1 can act as a transcription silencer of the MLP when it forms a complex with adenovirus L4-22K protein through the low-affinity binding site at region 1, located near the downstream element (DE) [113].

Lastly, E1A protein is proposed to inhibit the host immune response during adenoviral infection. Under inflammatory conditions, CBP/p300, Sp1, and the signal transducer and activator of transcription 1 (STAT1) dimer form a complex that increases transcription of the intercellular adhesion molecule-1 (ICAM-1) gene, which acts as an immune system stimulator. During adenoviral infection, separate molecules of E1A bind to the STAT1 dimer and to CBP/p300, preventing the formation of the complex and displacing Sp1 protein to support infection (Figure 2E) [114].

Additionally, the adenoviral genome has 15 highly conserved quadruplexes across different species, of which the most conserved were three G4Qs along the length of the E2B gene [107]. It is likely they play a crucial role in transcriptional regulation with the use of Sp1 protein.

#### 3.2.7. Parvoviridae

The Parvoviridae family includes highly diverse ancient species of ssDNA viruses that are capable of infecting a broad range of hosts from arthropods to mammals. Several genera within the subfamily Parvoviridae can cause ranging severity of disease in humans.

##### Erythroparvoviridae

The most-studied autonomously replicating human pathogenic virus of the Parvoviridae family is parvovirus B19 (B19V), causing ‘fifth disease’ in children, typically a self-limiting rash, but it can also cause more serious complications in immunocompromised individuals. Although only several cases are registered per year, 2024 experienced a robust rebound outbreak post-COVID-19 pandemic [159]. B19V has a ssDNA genome of ~5.6 kb in length, and it possesses only one functional promoter, p6, which is transactivated by the viral nonstructural protein 1 (NS1). Although not fully elucidated, NS1 is suggested to access the promoter through binding to Sp1, which has three or four sites in the p6 promoter (Figure 2D) [115,116]. The same NS1–Sp1 complex is suggested to activate p21/WAF1 expression, a cell cycle regulator promoting G1 cell cycle arrest [116].

##### Dependoparviridae

Adeno-associated viruses (AAVs) cause lifelong infection that is typically asymptomatic. Their low pathogenicity and broad cellular permissiveness are what has attracted the attention of researchers and eventually made AAV vectors the first gene therapy. AVVs can only replicate within the presence of a helper virus, which is commonly various adenoviruses or herpesviruses [117].

AVVs have a ssDNA linear genome of ~4.7 kb in length, and they contain three promoters, p5, p19, and p40, with the former two encoding non-structural Rep proteins and the latter encoding structural viral proteins (VPs). It has been suggested that Rep proteins transactivate p19 and p40 promoters while decreasing activity of the p5 promoter via a peculiar mechanism. In AVV-2, p5 and p19 were shown to form an architectural complex between Rep and Sp1 bound to their respective promoters, where they form a loop bringing the sequences together (Figure 2F) [61]. Additionally, Rep proteins interact with a broad range of partners to ensure replication suppression and activation depending on the presence of helper virus.

Being a satellite virus, AAV profits from interactions with the machinery of other viruses. In vitro, Rep78 of AVV-2 can prevent TBP and Sp1 from accessing and activating p97 and p105 core promoters of HPV-16 and -18, respectively. Such transcriptional suppression of the HPV genome leads to decreases in cell transformation [117,118].

## 4. Sp1 Cellular Network Within the Context of Viral Infection

The notion of concurrent evolution of viruses and hosts implies that the two would not exist in their current forms without each other. To protect themselves from viral attacks, the hosts evolved and maintained protective mechanisms, like IFN-inducible proteins (IFI), to up- or downregulate cellular processes when a cell has been invaded. To overtake the cellular machinery, many viruses come equipped to evade those same processes, such as through immune evasion, immortalization, chronic inflammation, and dysregulation of energetics and epigenetics—all of which inherently benefit the virus [75]. As we discuss the opposing forces of virus and host, it is important to remember that, although we use language granting agency to both “teams”, the process of evolution is opportunistic in nature and is not operating to reach a specific goal, which makes these processes all the more fascinating. In this section, we outline some of the most commonly affected Sp1 pathways in the presence of viral infection, which are also heavily linked to carcinogenesis.

### 4.1. Sp1 and Cell Cycle

Sp1 and an axis of principal tumor suppressors, p53-p21-pRb, exhibit mutual regulation to ensure adequate cell cycle progression. The axis inhibits activation of E2F TF, which upregulates all major metabolic pathways, including cell division, controlling a great variety of genes including the transcription factors in the proliferation group like Sp4, Oct-1, Maz, and Sp1 itself [160].

Both HAdV and HPV can downregulate and upregulate p53 availability through different mechanisms depending on the stage of infection, presumably to prevent early apoptosis and later to employ cell death for progeny spread. EBV [161] and MCPyV [162] have only been reported to downregulate p53. Conversely, pRb is downregulated by HAdV, HPV, EBV, Kaposi’s sarcoma-associated herpesvirus (KSHV) [161], MCPyV [162], and HTLV-1 [163] to liberate E2F TF from its inhibition and upregulate all major metabolic pathways. This, in turn, leads to active cell division and an increase in viral gene synthesis [161].

### 4.2. Sp1 and Other TFs

Sp1 controls the levels of other transcription factors, like NF-kB [98] and MYC [11], which each have three Sp1bs in their promoters. NF-kB is a key player in the early response to pathogens; it is involved in inflammation, cell survival, proliferation, and therefore tumorigenesis. Viruses commonly hijack NF-kB pathways [79] to either upregulate them for pro-survival mechanisms and activation of viral expression or downregulate them for immune evasion [84]. Myc is similarly highly linked to all essential cellular activities, thereby acting as a master regulator TF. Myc is a ubiquitously expressed TF that contributes to all major metabolic pathways (glycolysis, pentose phosphate pathway, nucleotide synthesis, tricarboxylic acid cycle, etc.) [161].

Viruses like HSV-1, EBV, and KSHV both upregulate and downregulate the NF-κB pathway to avoid the immune system and enhance viral expression, respectively. Conversely, HIV-1 has only been reported to upregulate this pathway, while HBV has only been reported to downregulate the same pathway [84], possibly through their influences on Sp1. HAdV and HPV both act on the MYC promoter to upregulate its expression and also downregulate Myc copies, supposedly to aid in the latency of the infection [161]. Meanwhile, EBV translocates the MYC gene within the Ig heavy locus to increase promoter–enhancer interaction [161].

### 4.3. Sp1 and Genomic Stability

Human telomerase reverse transcriptase (hTERT) is a catalytic subunit of a ribonucleoprotein that elongates telomeres with each cell division. Around 75–85% of human cancers have aberrant expression of hTERT, as it plays a key role in achieving cell transformation and cellular senescence [164,165]. Viruses may trigger hTERT association with TFs, amplification, PTMs, and mutations. Sp1 has five binding sites in the hTERT promoter [138,166], controlling its expression along with Myc/Maz, p53, and NF-kB proteins.

HPV, KSHV [164], HCMV [95], HSV-1, and HTLV-1 [57,165] upregulate hTERT expression in a Sp1-dependent manner. HBV’s influence on hTERT remains unclear, with contradictory reports suggesting the virus is capable of both up- and downregulation of the promoter [165]. Remarkably, viruses like EBV and HTLV-1 regulate hTERT expression in accordance with their stage of infection. Shortly after initial infection, EBV upregulation of hTERT enforces latency, while downregulation can trigger a lytic cycle [164].

Additionally, in tumor tissues, HBV and HPV genomes are often found to be integrated within the hTERT promoter, while EBV, KSHV, and some other genomes of herpesviruses are integrated within telomeres [164,165]. Nonetheless, a canonical tumor MCPyV is not associated with hTERT upregulation, and MCPyV tumors rarely have activated hTERT [162]. However, a closely related Lyon-IARC virus was reported to activate hTERT via Sp1 [167]. Lastly, HIV-1 reduces hTERT activity, shortening telomeres, which supposedly contributes to immune system decline [168].

## 5. Epigenetics in Viral Infections

Viral infections manipulate a multitude of cellular mechanisms for their benefit, and cell epigenetics is no exception. Epigenetic modifications within the context of viral infection can aid in temporal regulation of the viral cycle and hinder cell immune responses. These modifications allow for context-specific gene expression and can be implemented in three ways: direct DNA methylation of CpG (5′–C–phosphate–G–3′) islands, histone modifications, and utilization of non-coding RNAs [75].

The first to be discovered and the most common epigenetic modification in eukaryotes is the methylation of CpG promoter islands executed by DNA methyltransferases (DNMT), which acts as a fundamental silencing mechanism. Methylation promotes closed chromatin conformation and downregulates expression by creating steric hindrance, changing the shape of the DNA twists, and impacting nucleosome positioning or attracting repressive proteins that restrict PolII access (see below). Recent reports argue that it is the lack of TF binding that triggers the methylation to occur. CpG island methylation within Sp1bs was shown to downregulate gene expression in cancer [169] and viral infections [60,169,170].

Eukaryotic DNA is packaged into chromatin, which is classically described as “beads on a string”. Twisted DNA is wrapped around nucleosomes, a basic unit of chromatin made of the canonical histones H2A, H2B, H3, H4, and an additional linker histone, H1. In the case of DNA virus infections, the foreign viral genome is also wrapped around histones that are either virally encoded or more often belong to the host. Chromatin remodeling is achieved via post-translational modifications of the histone proteins, which either lower or enhance the protein’s affinity towards nucleic acids, which subsequently impacts DNA accessibility and therefore protein expression [75].

In this section, we provide examples of epigenetic modifications associated with DNA- or retroviruses in the context of Sp1 protein when the target of the modification is the virus or the cell.

### 5.1. Epigenetic Modifications: When the Target Is the Virus

Viruses use methylation dynamics to manage the expression of genes from their regulatory regions. Methylation prevents viral protein binding, like E2 (HPV) or LTag (BKpyV), and host Sp1 (and other TFs) to the control region, which downregulates viral replication. Similarly, binding of said factors can prevent methylation at these sites [54,171].

HBV genome methylation status has also been linked to viral infection and virus-associated cancer, hepatocellular carcinoma (HCC). Upon HBV infection, DNMT expression is upregulated; however, it is unclear whether that aids or hinders viral infection. In support of the second hypothesis, methylation of the epigenome, cccDNA, is recognized to be a defense mechanism leading to downregulation of viral expression. During HCC, a specific CpG positioned in the pre-Core promoter, which controls pregenomic RNA production and overlaps with the carcinogenic X gene, is not methylated [172]. Polyomaviral NCCR lacks methylation, hinting at a conserved evolutionary mechanism. It is hypothesized that multiple Sp1bs of the NCCR prevent viral methylation by the host cell [56].

EBV is an exceptional case of cycle control via methylation, with one active and four latent infection states (0, I, II, and III) determined by the methylation status of the promoters. This elaborate strategy of selective viral expression allows EBV to invade the immune system so well that it has infected over 90% of the adult population worldwide. For example, promoters of two viral transactivators, Zta and Rta, are only non-methylated during the lytic infection. However, a promoter may be silent despite its unmethylated Sp1bs, like Qp in latency III state, controlled by another mechanism. Remarkably, methylation is required for EBV reactivation. Viral transactivator BZLF1 preferentially interacts with methylated bindings sites (meZRE) on key viral promoters [173].

### 5.2. Epigenetic Modifications: When the Target Is the Cell

As described above, despite vast divergence within the viral “kingdom”, different agents target similar pathways within the host cell as an example of convergent evolution. Commonly, viruses modify DNMT activity to prevent methylation of their control regions while inducing methylation of the host genes to prevent Sp1 and other TF binding. Viruses including EBV, HBV, HIV-1, HPV, and KSHV promote hypermethylation of host immune response genes [174]. DNMT-driven hypermethylation of host genes also impacts factors controlling the cell cycle like RASSF1A in HBV infection [83], TGF-β receptor type 2 in KSHV infection [75], or p53 in HPV infection. For p53 specifically, this hypermethylation blocks the complex formation with Sp1 that is otherwise required for transcriptional activation [75]. Expression of other host genes also benefits the infection; for example, HCMV inhibits histone deacetylases (HDAC) 1 and 2, leading to chromatin opening and therefore enabling Sp1 binding [95].

## 6. Sp1 Post-Translational Modifications

Outside of the direct modifications on DNA, post-translational modifications (PTM) can occur on proteins, which can directly impact their functioning by altering a protein’s structure and binding affinity. This can largely expand the protein diversity required for organismal complexity and responses to stimuli [175]. Out of Sp1’s 785 amino acids, 164 residues (21%) are Ser or Thr; therefore, the Sp1 is highly phosphorylated and O-glycosylated. Sp1-associated PTMs alter transcriptional activation, cell growth and cycle, and DNA repair (a full list of the phosphorylated residues is available at: https://www.phosphosite.org/proteinAction.action?id=1312 (accessed 14 October 2024) [173]. The importance of PTM of Sp1 in cancer has been extensively reviewed elsewhere [175,176]. For the purposes of this article, we will only highlight the most-studied PTM, Sp1 phosphorylation in the context of viral infection.

One of the kinases commonly phosphorylating Sp1 is ataxia–telangiectasia-mutated (ATM) kinase that is responsive to DNA damage like viral replication and integration [102]. ATM phosphorylates Sp1 at Ser-131 during EBV infection, leading to Sp1-dependent viral protein localization to replication compartments [102]. Sp1 is also phosphorylated by ATM in HSV-1 infection at Ser-56/Ser-101; however, it appears to have no impact on the expression of viral or host Sp1-dependent genes [93].

During HIV-1 infection, Tat promotes Sp1 phosphorylation via the protein kinase DNA-activated catalytic subunit (PRKDC), contributing to viral expression from the LTR [69]. Simultaneously, Tat also prevents dephosphorylation of Sp1 by TCF4 in astrocytes specifically [70]. Sp1 phosphorylation in HBV infection is also speculated to contribute to viral expression; while the link between PRKDC and Sp1 is currently absent, PRKDC was shown to be a central regulator of HBV cccDNA expression [177], which is also dependent on Sp1 [77].

## 7. Authors Opinion: Sp1 as a Therapeutic Target

Given the highly essential nature of Sp1 for viral life cycles, the question arises regarding whether this reliance can be capitalized upon to create a therapy. In the cancer field, a number of molecules have been identified to either limit the Sp1 protein pool or limit Sp1-binding site availability and therefore lower Sp1-directed transcription.

For example, nonsteroidal anti-inflammatory agents (i.e., cyclo-oxygenase 2 inhibitors and related agents) promote the ubiquitination of Sp family members and target them for proteasome-dependent degradation. Meanwhile, anthracyclines, Mithramycin A, and analogs (mithralogs) bind and block Sp1bs and other GC-boxes. These antibiotics are well-described cancer therapeutics and can exhibit specificity towards a particular set of cell genes depending on the compound. Additionally, somewhat unconventional approaches were developed, like GC-rich oligomers acting as decoys binding Sp1 molecules [178] or Sp1 siRNAs [179]. Although both approaches limit Sp1 availability, they are hindered by the cellular uptake and immune system reaction. Lastly, many natural remedies are Sp1 inhibitors; for example, retinoid and zinc chelating compounds target Sp1 for caspase-dependent degradation. The latter also sequesters zinc in the environment, enhancing protein degradation. Curcumin, betulinic acid [180], and some cannabinoids [181] decrease miRNA-27a, reducing Sp1 expression, and they target Sp1 for proteasome-dependent degradation [170,182].

Mithralogs also showed potency against HIV-1 [73] and HSV-1 [74] through inhibition of viral transcription activation. Whether such a strategy could be employed in the setting of viral infections, even perhaps temporarily to tip the scales in favor of host survival, is yet to be determined. Considering that, despite the momentum mithralogs have gained as promising therapeutics for some cancers, they and other Sp1 inhibitors remain a relatively un-applied class of drugs due to high toxicity. Adverse side effects are unavoidable when targeting host protein or its binding site, which is the largest set back of an omnipresent target like Sp1.

As an alternative, several juxtapositioned Sp1 binding sites can form a G-quadruplex secondary structure that acts as a regulatory element that can be found within viral genomes and are integral to viral transcription [47,183]. Therefore, mithralogs and other G4Q ligands are possible avenues for drug development [74,184]. Yet, they would have to circumvent a similar off-target problem, as G4Qs of different conformations are widely found in the human genome [14,38,182]

Overall, careful consideration of Sp1 significance within the life cycle of a specific virus, along with the toxicity and unique effects of each compound, must be taken into account to evaluate the therapeutic potential of a novel drug.

## Figures and Tables

**Figure 1 viruses-17-00295-f001:**
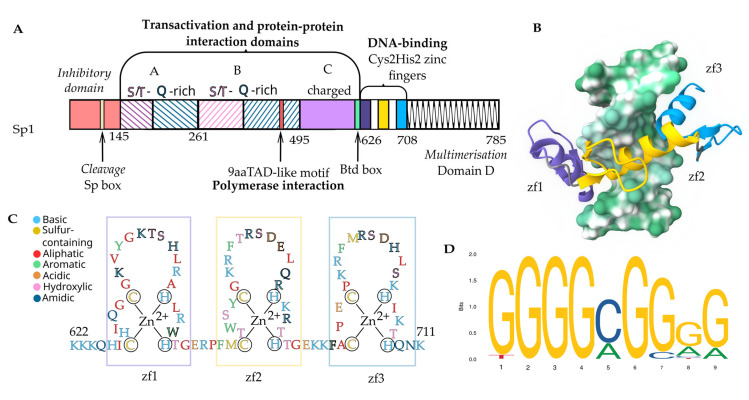
Sp1 structure and function. (**A**) Schematic of Sp1 primary structure. Notable domains and motifs; functions confirmed by direct experimental evidence are bolded, functions hypothesized from indirect evidence are italicized. (**B**) Proposed 3D model of interaction of Sp1 zinc fingers and duplex DNA. (**C**) Residues of Sp1 zinc fingers color-coded by the type of amino acid, (**D**) Sp1 DNA binding motif, JASPAR MA0079.5 [10,19,20,21,22].

**Figure 2 viruses-17-00295-f002:**
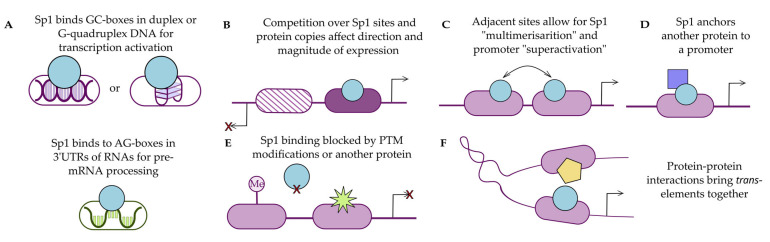
Sp1’s activities as a transcription factor and beyond. (**A**) Top: Sp1 binding to DNA of different conformations in promoter regions to facilitate polymerase assembly; bottom: Sp1 binds to RNA to facilitate pre-mRNA stability during processing. (**B**) Binding site of lower affinity (left) and site of stronger affinity (right) compete for available Sp1 copies to define transcription direction. (**C**) Hypothesized Sp1 oligomerization leading to enhanced transcriptional activation in juxtapositioned Sp1bs. (**D**) Sp1 using DNA-binding domain to grant another protein access to the promoter. (**E**) Sp1 bindings sites blocked via post-translational modifications (PTMs) and subsequent chromatin rearrangements or by another protein. (**F**) Sp1 interaction with another protein strengthens polymerase assembly or supposedly brings polymerase complex from another promoter.

**Figure 3 viruses-17-00295-f003:**
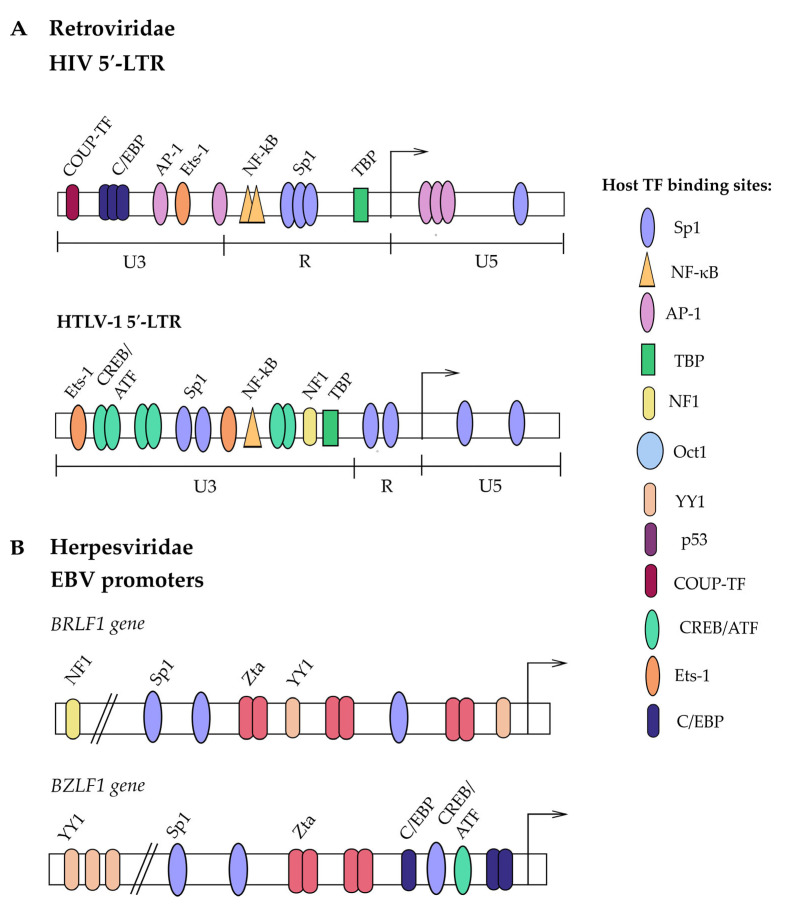
Sp1 binding sites and other TFs in promoters of Retroviridae (**A**) and Herpesviridae (**B**). Schematics highlight TFs based on their relevance to Sp1 and viral life cycle. Please, refer to the literature to understand the full complexity of the viral promoters [17,121,122].

**Figure 4 viruses-17-00295-f004:**
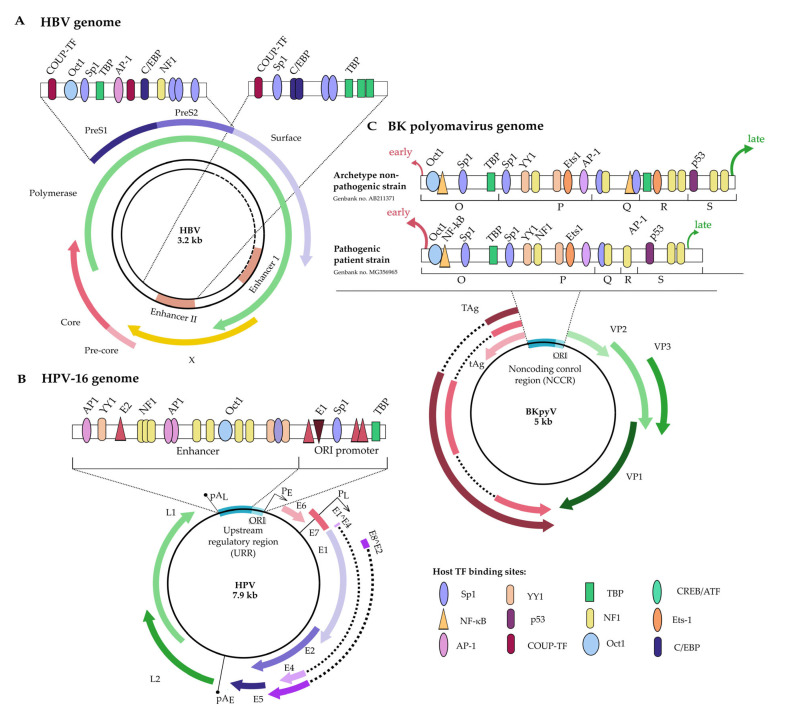
Sp1 binding sites and other TFs in genomes of Hepadnaviridae (**A**), Papillomaviridae (**B**), and Polyomaviridae (**C**). Schematics highlight TFs based on their relevance to Sp1 and viral life cycle. Please, refer to the literature to understand the full complexity of the viral promoters [47,49,105,134,136,137].

**Table 1 viruses-17-00295-t001:** Brief description of mechanisms involving Sp1 protein of DNA and retroviruses discussed in this article.

Family	Virus	Mechanism Description
Retroviridae	HIV-1	5′LTR includes three Sp1bs that comprise four G4Q structures, integral to viral expression ○ Sp1bs of LTR shown to ↑ and ↓ promoter expression through mutagenesis [39]. Correlation between nucleotide variants and disease progression in patient samples reported (e.g., Sp1-III C to T mutation → disease severity ↑) [63,64,65] ○ Mutations of Sp1bs in 5′LTR → stochastic changes in transcriptional phenotype of infected cells → latent viral reservoir ↓ [66,67] ○ In microglia, Sp3 protein blocks Sp1bs → transcription ↓ [62] ○ Mutagenesis of Sp1bs downstream of TSS → no difference in expression [68] ○ Sp1 is phosphorylated by PRKDC [69], Tat prevents Sp1 dephosphorylation by TCF4 (in astrocytes [70]) → viral expression ↑ Sp1 recruits to its binding sites HAT and HDAC → viral transcription ↑↓ ○ TRIM5α, innate immune factor, and Sp1 recruit HDAC to 5′LTR → reverse transcription ↓ [71] ○ Sp1 anchors CTIP2 and LSD1 → to histone deacetylation (via HDAC) and demethylation, respectively → viral expression ↓ [72] ○ In microglia, Sp1 anchors NF-IL6 and COUP-T in the NF-κB promoter → Tat expression ↑ → viral transcription ↑ [18] IFI16, an antiviral factor, sequesters Sp1 protein → viral replication ↓ [73] Mithralogs shown potency → viral transcription ↓ [74]
	HTLV-1	The HTLV-1 genome has six Sp1 binding sites positioned in sets of two in the U5, R, and U3 regions of each LTR [45] → Sp1bs control both sense (activation) and antisense (latency) strands ○ HBZ–JunD–Sp1 complex activates expression of HBZ by binding at 3′LTR → viral latency ↑; expression of hTERT → promoter cell immortality ↑ [57] ○ HBZ displaces CREB-2 at U5 → Sp1 protein and PolII binding ↓ → viral replication ↓ [57] ○ Sp1bs in 5′LTR are hypermethylated when integrated [75], while 3′LTR remains intact in all ATL cases → viral latency ↑ [57]
	HERV-K	Sp1 is present in LTR of HERV-K viruses and is likely integral to HERV-K propagation since present in undifferentiated cells [55] Sp1 overexpression in cancers → HERV-K promoter demethylation ↑ → viral proteins ↑ → pathogenicity ↑ [76]
Hepadnaviridae	HBV	Sp1bs are distributed: one in S1 promoter, three in PreS/S2, two in PreC/Core promoter (form a G4Q structure, integral to viral transcription) [38,47,77] ○ KLF15bs and Sp1bs overlap in PreCore/C and EnII promoters → synergistic or displacing interaction → viral expression ↑↓ [78] ○ NF-κB prevents Sp1 binding at preC/C and EnII → viral expression ↓ [79] HBx protein leads to HCC development ○ HBx-HOTAIR lncRNA complex binds to Sp1 promoter → Sp1-led transcription ↑ → cancer progression ↑ [78] ○ HBx–HDAC1 complex deacetylates Sp1 protein → DNA binding affinity ↓ → Sp1-led transcription ↓ [80] ○ HBx-DNMT complex increases Sp1bs methylation → cell cycle control proteins ↓ (e.g., RASSF1A [78] & CDKs) → cancer progression ↑ [75,81] ○ HBx downregulates NF-κB → immune system recognition ↓ → Sp1 availability ↑ → expression of immortality-related genes (e.g., hTERT [82] and DKK1 [77]) → cancer progression ↑ [80,83] ○ Hbx also upregulated NF-κB → activates pro-survival genes ↑ and immune surveillance → cancer progression ↓ [84]
Papillomaviridae	HPV	Sp1 and E2 control expression direction and infection state via an antagonistic relationship based on steric hindrance, differential affinity, and methylation patterns of binding sites [85] ○ Cell differentiation → Sp1bs affinity ↑ [86] and nucleosome displacement → viral transcription ↑ [59] ○ DNA methylation of the proximal vs. distal E2bs ↑ → cancer progression ↑ [85] ○ E2 gene loss → lack of an antagonistic relationship between E2 and Sp1 → E6/E7 expression dysregulation → cancer progression ↑ [48] ○ Sp1bs are present in all alpha-HPVs, motifs may be GC/GA or GT [33]
Herpesviridae	HSV-1	Sp1 is essential for IE gene expression (e.g., ICP0, ICP4, and ICP27) in absence of viral transactivators (e.g., early infection [87,88], latent stage, or stress [45]) ○ KLF15 binds Sp1bs within IE genes promoters → viral transcription ↑↓ [89] Sp1 participates in transcriptional activation of leaky-late genes ○ In VP5, Sp1bs are indispensable for expression [88,90] ○ In VP16, Sp1bs before the TATA box are redundant [91,92], but Sp1bs within the gene form a G4Q that binds Sp1 and Erg-1 ○ Erg-1 displaces activatory Sp1 and brings NAB2 (→ HDAC) → chromatin closure → viral latency ↑ [58] Sp1 is phosphorylated by ATM → no difference in expression [93] Mithralogs shown potency → viral transcription ↓ [74]
	HCMV	MEI promoter contains seven Sp1bs, at least two in proximal enhancer are redundant [94] ○ Sp1 interacts with IE72 and IE85 → viral expression ↑ [95] ○ Sp1 anchors IE72 at MEIP → viral latency ↑ [96] CMV virion binds to gB and hB cell receptors → Sp1 expression ↑ [97,98]
	EBV	Zp and Rp contain two and three Sp1bs, respectively ○ Some genes (e.g., Zta) lack Rtabs → Sp1–MCAF1–Rta complex activates transcription [60] ○ p53–Sp1 complex binds to Zp → viral transcription ↑ [99] ○ Zp only requires Sp1 for full vs. leaky expression [100] ○ ATM kinase phosphorylates Sp1 [97] → Sp1 removes HDAC2 from Sp1bs in Zp [101] → viral protein localization to replication compartments [102]
Polyomaviridae	BKPyV	Multiple Sp1bs, found in NCCR of each polyomavirus; Sp1bs control expression direction and infection state [103,104,105,106] ○ In archetype strain, Sp1bs proximal to late genes have higher affinity → viral latency [103] ○ In patient strain, deletion of that Sp1bs → early expression ↑ ○ In laboratory strain, deletion of Sp1bs next to early gene start → abrogation of viral replication [49] Sp1bs protect NCCR from host methylation [56] → viral expression ↑
Poxviridae	VACV	Virion never enters the nucleus; replication takes place in inclusion bodies [107] Sp1 associates with viral genome; Sp1 functionality unconfirmed [108]
Adenoviridae	HAdV	Sp1bs are found in ITRs of HAdV-4/-7,-40,-41, -52 species [109,110] Sp1 is an activator of E1B [111] and IX [112] promoters in AdVs-2/-5 Sp1–Maz complex activates MLP in AdV-4/-7 [113] Sp1–L4-22K complex binds to DE → viral transcription ↓ [113] E1A molecules separately bind to CBP/p300 and STAT1 protein → prevent CBP/p300-Sp1–STAT1 complex formation → ICAM-1 expression ↓ → immune surveillance ↓ [114]
Parvoviridae	B19V	NS1-Sp1 complex activates p6 promoter [115,116] and activates p21/WAF1 expression, a cell cycle regulator promoting G1 cell cycle arrest [116]
	AVV	Rep-Sp1 complex bridge together p5 and p19 while bound, respectively [61] Rep78 of AVV-2 can prevent Sp1 from activating p97 and p105 promoter of HPV-16/-18, respectively → HPV transcription ↓ → cancer progression ↓ [117,118]

↑—increased; ↓—decreased; ↑↓—can both increase and decrease; →—“which leads to”.

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
