# Peer review of "Viral Appropriation of Specificity Protein 1 (Sp1): The Role of Sp1 in Human Retro- and DNA Viruses in Promoter Activation and Beyond"

_viruses, 2025, doi:10.3390/v17030295_

Round 1

Reviewer 1 Report

Comments and Suggestions for Authors

The authors wrote a review about the transcription factor specific protein 1 (SP1) and how human retro and DNA viruses use this TF for transcription initiation of the viral promoter. It is a thorough review and details

  • the background of SP, incl. evolutionary conservation, protein structure, DNA binding sites, mechanisms of transcriptional activation.
  • The SP1 binding site in the promoter of different human retro and DNA viruses are discussed, including a brief overview of the pathogenesis of each virus, a description of the viral promoter and SP1 binding sites, and the role SP1 plays in transcription such as the interaction with other TFs where relevant. For the human retroviruses, HIV, HTLV and HERVs are discussed and for the human DNA viruses, HepB, HPVs, herpes viruses (HSV-1, CMV, EBV), polyomaviruses, poxviruses (variolas and Mpox), Adenovirus, parvovirus B19 and AAVs.
  • Then finally, the role of SP1 beyond direct binding to the SP1 binding site is discussed, such as the effects of SP1 on the cell cycle during viral infection and possible subsequent oncogenesis and epigenetic regulation of viral DNA.

Overall, the review is well written, easy to follow and understand. The literature is well reverenced and summarized. The figures are clear and support the main text. Except for fig4, which appeared as a black square and was unintelligible, and accompanied by the wrong legend.

I only have minor comments that I hope will help improve an already well written manuscript.

  • Perhaps a short summary of the promoter of SP1 itself can be added to the background information of SP1 to indicate how expression of the TF is regulated.
  • For the manuscript organization, the human retroviruses are listed in paragraph 3.1 and introduced with a small summary as to what classifies a retrovirus followed by a specific paragraph for each genus that infects humans (lenti 3.1.1, delta 3.1.2 and beta 3.1.3). I think the DNA viruses could be introduced similarly with a small introduction followed by the different viruses in subparagraphs. Now the list of viruses continues without introducing the transition from retro to DNA virus.
  • The section on the herpesviruses missed a description of HSV-2, VZV, HHV6, 7 and 8. If there are no reports available then this could be mentioned. However, some quick searching resulted in some interesting studies of SP1 in VZV (PMID: 18815296 and PMID: 10989187).
  • Similarly, HIV-2 and HTLV-2 could be included in the lentivirus and delta retrovirus sections, preferably with a schematic of the promoter organization.
  • The use of the word ‘appropriation’, gives the virus agency which it does not have, and is in my opinion not appropriate for a description of a virus that is opportunistic in nature. This extends to the title and lines 35-36, and ‘many viruses order SP1 to do their bidding.’ Line 36. Viruses do not order proteins around.
  • Line 46-47: ‘The Sp1-4 proteins are further separated from the other five members by the presence of the glutamine-rich regions (Fig.1A).’ Not clear who the other five members are. A few sentences before, SP1 was introduced as being part of a family of 26 members.
  • Line 143: ‘Yet, Sp1 also activates cellular transcription, including that of immune genes, and accordingly,…..’ It could be interesting to give some examples of immune genes that are regulated by SP1.
  • Line 187-188: ‘HIV-1 viral life cycle starts with genome integration into the host DNA’. Arguably, HIV-1's cycle starts with the process of reverse transcription.
  • Line 215: ‘…to increase NF-κB levels leading to Tax expression…’ HIV-1 has Tat protein, not Tax.
  • The section of HTLV-1 describes the tax/rex feedback loop, the HIV section could benefit from a description of the Tat/TAR feedback look. SP1+TBP increase stochastic transcriptional bursting of the HIV promoter, leading to increased Tat expression (reviewed in PMID: 37766375)
  • Lines: 221-223: ‘HIV-1 infected cells have an ability to switch between latent and acute phenotypes which contributes to the maintenance of the latent viral reservoir – an integral part of infection persistence.’ This is incorrect. The cell does not switch between two states. The HIV promoter can switch between two states and this has an effect of the host cell’s phenotype.
  • Lines: 224-226: ‘When Sp1 binding sites are mutated, the Tat-mediated positive-feedback loop fails to be established leading to stochastic changes of transcriptional phenotype in infected cells, resulting in lack of a latent viral reservoir [66,67]’. This is debatable. I would argue that without the SP1bs the virus is defective and cannot replicate, as such, it is a dead-end product and there is thus lack of virus replication, (and therefore there is also no latent provirus).
  • Line 273: ‘…Tax re-activation.’ Better phrased as Tax expression or virus re-activation. The protein itself was not dormant/latent and in need of re-activation.
  • Lines 334-336: ‘HBx also interacts with long non-coding RNA, like lncRNA homeobox (HOX) transcript antisense RNA (HOTAIR) that has a positive binding site on the Sp1 gene promoter, leading to …’ Would benefit from stating that SP1 transcription and expression is enhanced, which then leads to enhanced expression of genes with an SP1bs.
  • Line 346: ‘Papillomaviruses (PVs) are circular dsDNA viruses infecting a wide range of hosts, ~8 kb in length [94].’ Would benefit from rephrasing.
  • Lines 407-408: ‘Herpes simplex virus -1 (HSV-1) infects nasal, ocular, oral cavities and may manifest in sensory neurons leading to encephalitis, blindness and neonatal deformities [54].’ This could benefit from a more nuanced description. HSV-1 has tropism for epithelial cells and sensory neurons, in the latter it establishes latent infection. Only in rare cases leading to encephalitis or infection of the optical nerve leading to acute retinal necrosis (ARN) causing blindness. HSV-1 can be transmitted congenitally or upon delivery of newborns. Very rare, but potentially devastating.
  • Line 455: ‘..is suggested to require Sp1 to maintain genome in latency (Fig.2D) [115].’ Would benefit from rephrasing.
  • Line 504: ‘…– such sequences are not in nature (Fig.4C) [48].’ Would benefit from rephrasing. For example, ‘have not been observed/detected in clinical isolates’.

Author Response

Thank you for taking the time to review our manuscript.

Kindly see the responses to your comments/suggestions below.

Reviewer 1:

The authors wrote a review about the transcription factor specific protein 1 (SP1) and

how human retro and DNA viruses use this TF for transcription initiation of the viral

promoter. It is a thorough review and details the background of SP, incl. evolutionary conservation, protein structure, DNA binding sites, mechanisms of transcriptional activation.

The SP1 binding site in the promoter of different human retro and DNA viruses are

discussed, including a brief overview of the pathogenesis of each virus, a description of

the viral promoter and SP1 binding sites, and the role SP1 plays in transcription such as

the interaction with other TFs where relevant. For the human retroviruses, HIV, HTLV

and HERVs are discussed and for the human DNA viruses, HepB, HPVs, herpes

viruses (HSV-1, CMV, EBV), polyomaviruses, poxviruses (variolas and Mpox),

Adenovirus, parvovirus B19 and AAVs.

Then finally, the role of SP1 beyond direct binding to the SP1 binding site is discussed,

such as the effects of SP1 on the cell cycle during viral infection and possible

subsequent oncogenesis and epigenetic regulation of viral DNA.

Overall, the review is well written, easy to follow and understand. The literature is well

reverenced and summarized. The figures are clear and support the main text. Except

for fig4, which appeared as a black square and was unintelligible, and accompanied by

the wrong legend.

I only have minor comments that I hope will help improve an already well written manuscript.

Perhaps a short summary of the promoter of SP1 itself can be added to the

background information of SP1 to indicate how expression of the TF is regulated.

--Added a sentence to highlight Sp1’s control “The Sp1 promoter itself is directly regulated by other transcription factors like Myc, Maz and HIF, and is autoregulated through several Sp1 binding sites.” (Line 35)

For the manuscript organization, the human retroviruses are listed in paragraph 3.1 and introduced with a small summary as to what classifies a retrovirus followed by a specific paragraph for each genus that infects humans (lenti 3.1.1, delta 3.1.2 and beta 3.1.3). I think the DNA viruses could be introduced similarly with a small introduction followed by the different viruses in subparagraphs. Now the list of viruses continues without introducing the transition from retro to DNA virus.

--Good point. We have added the heading “DNA viruses” and renumbered our DNA viruses heading to reflect them as a sub-category of that heading.

The section on the herpesviruses missed a description of HSV-2, VZV, HHV6, 7 and 8. If there are no reports available then this could be mentioned. However, some quick searching resulted in some interesting studies of SP1 in VZV (PMID: 18815296 and PMID: 10989187).

Similarly, HIV-2 and HTLV-2 could be included in the lentivirus and delta retrovirus sections, preferably with a schematic of the promoter organization.

--Yes, as the reviewer has pointed out there are other viruses within the families described that are relevant to the topic and deserving of attention. Unfortunately, at the time of initial submission, the article was already over 9000 words (excluding the article backend). And therefore, we made the decision to highlight one virus from each of the subfamilies included that has relevance to human virology.

The use of the word ‘appropriation’, gives the virus agency which it does not have, and is in my opinion not appropriate for a description of a virus that is opportunistic in nature. This extends to the title and lines 35-36, and ‘many viruses order SP1 to do their bidding.’

Line 36. Viruses do not order proteins around.

--This is a valid point. We have modified Line 36 to state “…use Sp1 for their own transcriptional activation” (now Line 38). We also offer up “Viral exploitation of Sp1…” as an alternative title

Line 46-47: ‘The Sp1-4 proteins are further separated from the other five members by the presence of the glutamine-rich regions (Fig.1A).’ Not clear who the other five members are. A few sentences before, SP1 was introduced as being part of a family of 26 members.

--We have added “…separated from the other five Sp1-like protein members…” for clarification (Line 48)

Line 143: ‘Yet, Sp1 also activates cellular transcription, including that of immune genes, and accordingly,…..’ It could be interesting to give some examples of immune genes that are regulated by SP1.

--We have added a few examples to the immune genes comments “like those involved in the RIG-I pathway” (Line 147)

Line 187-188: ‘HIV-1 viral life cycle starts with genome integration into the host

DNA’. Arguably, HIV-1's cycle starts with the process of reverse transcription.

--We have modified this section to elaborate and more properly highlight this (and a similar comment by another reviewer) (Line 192-196)

Line 215: ‘…to increase NF-κB levels leading to Tax expression…’ HIV-1 has Tat protein, not Tax.

--fixed (now Line 227)

The section of HTLV-1 describes the tax/rex feedback loop, the HIV section could benefit from a description of the Tat/TAR feedback loop. SP1+TBP increase stochastic transcriptional bursting of the HIV promoter, leading to increased Tat expression (reviewed in PMID: 37766375)

--We have included a few extra details and clarifications on this and have this article cited under in the paragraph on maintenance of the viral reservoir (Line 235-241).

Lines: 221-223: ‘HIV-1 infected cells have an ability to switch between latent and acute phenotypes which contributes to the maintenance of the latent viral reservoir – an integral part of infection persistence.’ This is incorrect. The cell does not switch between two states. The HIV promoter can switch between two states and this has an effect of the host cell’s phenotype.

--Reworded to “HIV-1 LTR promoters have the ability to switch between silent and activates state which contributes to the maintenance of the latent viral reservoir…“ (now Line 235)

Lines: 224-226: ‘When Sp1 binding sites are mutated, the Tat-mediated positive-feedback loop fails to be established leading to stochastic changes of transcriptional phenotype in infected cells, resulting in lack of a latent viral reservoir [66,67]’. This is debatable. I would argue that without the SP1bs the virus is defective and cannot replicate, as such, it is a dead-end product and there is thus lack of virus replication, (and therefore there is also no latent provirus).

--We have reworded to “…resulting in poor establishment of infection” (Line 241)

Line 273: ‘…Tax re-activation.’ Better phrased as Tax expression or virus re-activation. The protein itself was not dormant/latent and in need of re-activation.

--Modified phrasing to better describe this. (now Line 286)

Lines 334-336: ‘HBx also interacts with long non-coding RNA, like lncRNA homeobox (HOX) transcript antisense RNA (HOTAIR) that has a positive binding site on the Sp1 gene promoter, leading to …’ Would benefit from stating that SP1 transcription and expression is enhanced, which then leads to enhanced expression of genes with an SP1bs.

-- Rephrased to “HBx enhances Sp1 expression and by extension the expression its own genes through an interaction with long non-coding RNA, like lncRNA homeobox (HOX) transcript antisense RNA (HOTAIR) that has a positive binding site on the Sp1 gene promoter” (Line 348-350)

Line 346: ‘Papillomaviruses (PVs) are circular dsDNA viruses infecting a wide range of hosts, ~8 kb in length [94].’ Would benefit from rephrasing.

--Deleted “infecting a wide range of hosts” to keep with parallel format of the other virus descriptions.

Lines 407-408: ‘Herpes simplex virus -1 (HSV-1) infects nasal, ocular, oral cavities and may manifest in sensory neurons leading to encephalitis, blindness and neonatal deformities [54].’ This could benefit from a more nuanced description. HSV-1 has tropism for epithelial cells and sensory neurons, in the latter it establishes latent infection. Only in rare cases leading to encephalitis or infection of the optical nerve leading to acute retinal necrosis (ARN) causing

blindness. HSV-1 can be transmitted congenitally or upon delivery of newborns. Very rare, but potentially devastating.

--Rephrased “Herpes simplex virus -1 (HSV-1) has a tropism for epithelial and sensory neuronal cells, the latter in which it is able to establish latent infection [108] and may lead to encephalitis, blindness and neonatal deformities in rare cases”. (Lines 423-426)

Line 455: ‘..is suggested to require Sp1 to maintain genome in latency (Fig.2D) [115].’ Would benefit from rephrasing.

--Rephrased “…and it is proposed that it uses Sp1 to tether to its active site and maintain the genome in latency (Fig.2D)” (Line 473)

Line 504: ‘…– such sequences are not in nature (Fig.4C) [48].’ Would benefit

from rephrasing. For example, ‘have not been observed/detected in clinical

isolates’.

--Rephrased “such sequences have not been observed in clinical samples” (Line 523)

Reviewer 2 Report

Comments and Suggestions for Authors

Sviderskaia and Meier-Stephenson, Manuscript # 3477468 

This is a comprehensive and scholarly review on the role of the Sp1 transcription factor in promoter activation of human retroviruses and DNA viruses. The manuscript is well written and should be well received by the Viruses readership. I have largely stylistic and typographical comments, as well as changes that add clarity and accuracy. However, the authors may want to more carefully portray epigenetic mechanisms, as well as the relevance of viral nucleotide changes. Addressing of these comments are at the authors’ discretion.

Comments:

Line 53, perhaps add Sp1bs to the Abbreviation list

Line 65, should be “Sp1-led”

Line 80, “methylated GC-box”, the authors have not introduced transcriptional regulation by DNA methylation.

Line 83, which may also… stem… from evolutionary pressures 

Line 141, clarify: bind to the…viral…genome

Line 150, the standard abbreviation for Histone Acetyltransferase is HAT. It should be corrected throughout the text and in the abbreviation list.   

Line 156, commonly…exploited…by viruses 

Lines 188-190, “Upon integration, transcription responsive elements (TAR) anchors on the viral 5′ long terminal repeat (LTR), and brings transcription initiation factors together to express viral transactivator Tat protein.” As written, TAT is portrayed as a DNA-binding protein, as the LTRs are DNA sequences. These events should be more carefully described. After integration, cellular TFs facilitate low level HIV transcriptional activation that leads to TAT protein synthesis. When TAT levels accumulate, TAT binds to the nascent TAR RNA, with TAT recruiting transcription factors. These events result in a positive feedback loop, as high levels of TAT are then produced.

Lines 197-204, The authors should be more cautious (throughout) when discussing HIV nucleotide changes and their impact. The implications are that Sp1bs nucleotide changes affect viral fitness. Nucleotide variations are most informative when the changes become fixed and lead to a viral phenotype. The fitness of HIV subtypes can vary in terms of replication capacity and may be the result of numerous nucleotide differences. The authors may want to clarify which changes in Sp1bs unambiguously affect HIV replication. 

Nucleotide changes are based on a comparison to a reference sequence. The authors may want to use the term “nucleotide variations” which do not make any assumptions as to wild type versus mutant sequences. That said, reference 62 may describe site-directed mutagenesis experiments and those findings should be distinguished.  

Line 236,  genome contains two…LTRs…which

Lines 254-255, More careful wording is required. LTR sequences are identical, by definition. Also, Sp1bs and TATA boxes are palindromic. Stating that the 3’ LTR lacks a TATA box implies that the 5’LTR has a TATA box. Of course, the authors are describing the relevant antisense promoter at the 3’ end of the HTLV DNA genome. It might be better to say that the HTLV LTR can function as a 3’ -antisense promoter and lacks a ?functional? antisense TATA box.    

Line 306, that alters…the…epigenetic state 

Line 330, “HBx hypermethylates” should be changed to “HBx increases the activity of DNMTs”

Line 338, to deacetylate…the…Sp1 protein.   

Line 358, “along with Sp1 binding site”, should be either “the Sp1 binding site” or “Sp1 binding sites”.   

Line 370, …DNA… methylation at the proximal 

Line 378, executes…histone…acetylation and…DNA…methylation changing TF affinity.      

Line 476, (auto-)immune, change to autoimmune 

Figure 3 seems to have technical issues?

Lines 646-651, Introduce epigenetic principles earlier in article. DNA methylation is mentioned numerous times prior to line 646, but only here the concept of the CpG site methylation is described. 

Lines 655-657, It is stated that “methylation promotes closed chromatin conformation and downregulates expression through creating steric hindrance, changing the shape of DNA twists and impacting nucleosome positioning.” Although a credible mechanism, more commonly DNA methylation acts as a signal to recruit repressive proteins to a gene.

The authors could mention the histone code and DNA methylation modifications (mechanisms and consequences) earlier in the manuscript. In both case, modifications serve recruitment mechanisms for positive or negative transcriptional regulation.

Line 726, “Tat phosphorylates Sp1”.  Should say that Tat promotes phosphorylation.    

Author Response

Thank you for taking the time to review our manuscript.

Kindly see the responses to your comments/suggestions below.

Reviewer 2:

This is a comprehensive and scholarly review on the role of the Sp1 transcription factor

in promoter activation of human retroviruses and DNA viruses. The manuscript is well

written and should be well received by the Viruses readership. I have largely stylistic

and typographical comments, as well as changes that add clarity and accuracy.

However, the authors may want to more carefully portray epigenetic mechanisms, as

well as the relevance of viral nucleotide changes. Addressing of these comments are at

the authors’ discretion.

Comments:

Line 53, perhaps add Sp1bs to the Abbreviation list

--Added

Line 65, should be “Sp1-led”

--Fixed

Line 80, “methylated GC-box”, the authors have not introduced transcriptional regulation

by DNA methylation.

--A valid point. We now make reference to the later section so as to tie this in without too much deviation “(see Epigenetics in Viral Infections section)” (Line 83)

Line 83, which may also… stem… from evolutionary pressures

--Fixed 

Line 141, clarify: bind to the…viral…genome

--Fixed – “bind to viral genomes” (now Line 144)

Line 150, the standard abbreviation for Histone Acetyltransferase is HAT. It should be

corrected throughout the text and in the abbreviation list.  

--Fixed this and 2 additional uses. 

Line 156, commonly…exploited…by viruses

--Replaced  

Lines 188-190, “Upon integration, transcription responsive elements (TAR) anchors on

the viral 5′ long terminal repeat (LTR), and brings transcription initiation factors together

to express viral transactivator Tat protein.” As written, TAT is portrayed as a DNA-

binding protein, as the LTRs are DNA sequences. These events should be more

carefully described. After integration, cellular TFs facilitate low level HIV transcriptional

activation that leads to TAT protein synthesis. When TAT levels accumulate, TAT binds

to the nascent TAR RNA, with TAT recruiting transcription factors. These events result in

a positive feedback loop, as high levels of TAT are then produced.

--This paragraph has been revised (Lines 194-199)

Lines 197-204, The authors should be more cautious (throughout) when discussing HIV

nucleotide changes and their impact. The implications are that Sp1bs nucleotide

changes affect viral fitness. Nucleotide variations are most informative when the

changes become fixed and lead to a viral phenotype. The fitness of HIV subtypes can

vary in terms of replication capacity and may be the result of numerous nucleotide

differences. The authors may want to clarify which changes in Sp1bs unambiguously

affect HIV replication.

Nucleotide changes are based on a comparison to a reference sequence. The authors

may want to use the term “nucleotide variations” which do not make any assumptions

as to wild type versus mutant sequences. That said, reference 62 may describe site-

directed mutagenesis experiments and those findings should be distinguished. 

--Thank you and a valid point in distinguishing SDM studies and the viral fitness in the more global sense. We have reviewed the cases where this may have been overstated.

Line 236,  genome contains two…LTRs…which

--Fixed  

Lines 254-255, More careful wording is required. LTR sequences are identical, by

definition. Also, Sp1bs and TATA boxes are palindromic. Stating that the 3’ LTR lacks a

TATA box implies that the 5’LTR has a TATA box. Of course, the authors are describing

the relevant antisense promoter at the 3’ end of the HTLV DNA genome. It might be

better to say that the HTLV LTR can function as a 3’ -antisense promoter and lacks a

?functional? antisense TATA box.   

--Fixed  

Line 306, that alters…the…epigenetic state

--Fixed

Line 330, “HBx hypermethylates” should be changed to “HBx increases the activity of

DNMTs”

--Rephrased.

Line 338, to deacetylate…the…Sp1 protein.  

--Fixed

Line 358, “along with Sp1 binding site”, should be either “the Sp1 binding site” or “Sp1

binding sites”.  

--Fixed

Line 370, …DNA… methylation at the proximal

--Fixed

Line 378, executes…histone…acetylation and…DNA…methylation changing TF

affinity.     

--Fixed

Line 476, (auto-)immune, change to autoimmune

--Deleted “auto-” for clarity to capture both autoimmune and immune pathologies

Figure 3 seems to have technical issues?

--Our apologies. We will clarify the issue with the Journal what may have happened.

Lines 646-651, Introduce epigenetic principles earlier in article. DNA methylation is

mentioned numerous times prior to line 646, but only here the concept of the CpG site

methylation is described.

--Thank you and agreed. We considered shifting this section earlier in the manuscript in initial drafts but ultimately placed it where it currently is towards the bottom because it refers to numerous viral life cycles and processes that would need to be introduced first. We reasoned that epigenetic modifications are a more common concept as compared to the biology of specific viruses. We have now incorporated a line introducing this earlier (Line 152-154) as well as referring to the later Epigenetics section (Line 83 and 154). We hope that a reader can navigate through the article depending on their specific need and interest.

Lines 655-657, It is stated that “methylation promotes closed chromatin conformation

and downregulates expression through creating steric hindrance, changing the shape of

DNA twists and impacting nucleosome positioning.” Although a credible mechanism,

more commonly DNA methylation acts as a signal to recruit repressive proteins to a Gene.

--Incorporated the additional phrasing (Line 683)

The authors could mention the histone code and DNA methylation modifications

(mechanisms and consequences) earlier in the manuscript. In both case, modifications

serve recruitment mechanisms for positive or negative transcriptional regulation.

--We have added a line to highlight the concept earlier and make reference to the upcoming section for more detail (Line 152-154)

Line 726, “Tat phosphorylates Sp1”.  Should say that Tat promotes phosphorylation.   

--Fixed. 

Reviewer 3 Report

Comments and Suggestions for Authors

This review is a well-written and well-structured article that provides an in-depth analysis of the role of Sp1 in viral infections, covering both retroviruses and DNA viruses. The article is valuable and suitable for publication after addressing a few minor editorial issues.

  1. The black background reduces visibility, making it difficult for readers to discern details clearly. Consider adjusting the contrast or background color for better readability.
  2. The font size appears inconsistent with the rest of the text. Standardizing the formatting would enhance readability.
  3.  Including a table summarizing the roles of Sp1 in different viruses mentioned in the review would greatly enhance clarity, providing readers with a comprehensive overview of the key points discussed.

Overall, this review presents a thorough and insightful discussion, and with these minor revisions, it will be an excellent contribution to the field.

Author Response

Thank you for taking the time to review our manuscript.

Kindly see the responses to your comments/suggestions below.

Reviewer 3:

This review is a well-written and well-structured article that provides an in-depth

analysis of the role of Sp1 in viral infections, covering both retroviruses and DNA

viruses. The article is valuable and suitable for publication after addressing a few minor

editorial issues.

The black background reduces visibility, making it difficult for readers to discern details

clearly. Consider adjusting the contrast or background color for better readability.

The font size appears inconsistent with the rest of the text. Standardizing the formatting

would enhance readability.

--Thank you and apologies – I suspect there was an issue with the formatting transition during the submission process. We will look into the issue with the type-editors.

Including a table summarizing the roles of Sp1 in different viruses mentioned in the review would greatly enhance clarity, providing readers with a comprehensive overview of the key points discussed.

Overall, this review presents a thorough and insightful discussion, and with these minor

revisions, it will be an excellent contribution to the field.

--Thank you for your review. We have included a Table 1, as you have suggested to compile the roles of Sp1 by different viruses